# Current Knowledge about the Peritumoral Microenvironment in Glioblastoma

**DOI:** 10.3390/cancers15225460

**Published:** 2023-11-17

**Authors:** Gianluca Trevisi, Annunziato Mangiola

**Affiliations:** 1Department of Neurosciences, Imaging and Clinical Sciences, G. D’Annunzio University Chieti-Pescara, 66100 Chieti, Italy; annunziato.mangiola@unich.it; 2Neurosurgical Unit, Ospedale Spirito Santo, 65122 Pescara, Italy

**Keywords:** glioblastoma, glioma-associated microglia, glioma-associated stromal cells, immune infiltrate, microenvironment, peritumoral brain area, peritumor, tumor-infiltrating lymphocytes

## Abstract

**Simple Summary:**

In this manuscript, we review the most relevant biological findings on glioblastoma peritumoral tissue. This area, which is beyond frank tumoral tissue, presents a unique microenvironment with a peculiar cellular landscape and genomic and transcriptomic alterations that ultimately lead to glioblastoma progression or recurrence after treatment. Indeed, while some infiltrating tumor cells are found in about one third of cases, these alterations are seen even in non-infiltrated cases. This is of utmost translational interest, as the peritumoral brain zone (PBZ) is the real target of post-operative adjuvant treatment.

**Abstract:**

Glioblastoma is a deadly disease, with a mean overall survival of less than 2 years from diagnosis. Recurrence after gross total surgical resection and adjuvant chemo-radiotherapy almost invariably occurs within the so-called peritumoral brain zone (PBZ). The aim of this narrative review is to summarize the most relevant findings about the biological characteristics of the PBZ currently available in the medical literature. The PBZ presents several peculiar biological characteristics. The cellular landscape of this area is different from that of healthy brain tissue and is characterized by a mixture of cell types, including tumor cells (seen in about 30% of cases), angiogenesis-related endothelial cells, reactive astrocytes, glioma-associated microglia/macrophages (GAMs) with anti-inflammatory polarization, tumor-infiltrating lymphocytes (TILs) with an “exhausted” phenotype, and glioma-associated stromal cells (GASCs). From a genomic and transcriptomic point of view, compared with the tumor core and healthy brain tissue, the PBZ presents a “half-way” pattern with upregulation of genes related to angiogenesis, the extracellular matrix, and cellular senescence and with stemness features and downregulation in tumor suppressor genes. This review illustrates that the PBZ is a transition zone with a pre-malignant microenvironment that constitutes the base for GBM progression/recurrence. Understanding of the PBZ could be relevant to developing more effective treatments to prevent GBM development and recurrence.

## 1. Introduction

Glioblastoma (GBM) is the most common and most deadly primary brain tumor in humans, being classified as the fourth and most malignant grade of central nervous system (CNS) tumors by the World Health Organization (WHO) classification system [1]. Before 2016, gliomas were divided into different types and grades (from 1, the most indolent, to 4, the most aggressive) according to their histological characteristics. Classical histological characteristics of GBM include a diffusely infiltrative growth pattern, nuclear atypia, mitotic activity, increased cellular density, pseudopalisades, microvascular proliferation, and necrosis. The latter two were often considered as pathognomonic of GBM. However, in the last decade, molecular biology has figured strongly in the WHO CNS classification. Indeed, the new 2021 edition divided adult-type, diffuse gliomas into only three types: astrocytoma, IDH-mutant; oligodendroglioma, IDH-mutant and 1p/19q-codeleted; and glioblastoma, IDH-wildtype [2]. Therefore, all IDH-mutant diffuse astrocytic tumors are considered a single type (astrocytoma, IDH-mutant) and are then graded as CNS WHO grade 2, 3, or 4. Moreover, grading is no longer entirely histological, since the presence of CDKN2A/B homozygous deletion results in a CNS WHO grade of 4, even in the absence of microvascular proliferation or necrosis. Lastly, according to the 2021 classification, a diagnosis of glioblastoma should now be diagnosed in the setting of an IDH-wildtype diffuse and astrocytic glioma in adults if there is TERT promoter mutation or EGFR gene amplification or there are +7/−10 chromosome copy-number changes, regardless of histological features of a lower grading [2].

Gross total surgical resection (GTR) or maximal safe resection followed by radio-chemotherapy is considered the treatment of choice for newly diagnosed glioblastoma [3,4,5]. The extent of resection has been extensively associated with prolonged survival in these tumors [6,7].

However, the above strategies have prolonged the overall survival of our patients by only a few months, averaging 20 months from diagnosis in recent series. It is a common clinical experience to observe tumor recurrence in the area surrounding the surgical margins, generally between 1 and 2 cm from the resection cavity, even in cases of GTR confirmed by post-operative MR scans performed with the most-advanced machines and techniques and followed by full adjuvant therapy [8].

This has long been known in the neurosurgical community, with discouraging early experiences of recurrences even after hemispherectomies in a small series of patients in the 1920s [9]. Histopathological and biomolecular evidence of tumor cells located at relevant distances from the tumor core, both in the same hemisphere and in the contralateral one, led to the diffuse opinion that glioblastoma should be considered a systematic brain disease and not a delineated tumor [10,11].

Our interest in the “periphery” of GBM, namely, the brain tissue around MR contrast enhancement and seen as apparently normal at surgical resection, started with the initial observation of longer survival in GBM patients with no tumor cell infiltration of apparently normal white matter compared to patients with white matter infiltration (21 vs. 12 months) [12].

As our research went on, we realized that the alterations in the “periphery” were not limited only to the presence of frank tumor cells but also included a different cellular and molecular microenvironment compared to healthy brain tissue. This was true also in cases where no tumor cell infiltration was detected. Whether this is related to intracerebral tumor cell migration [13], to tumor-initiating stem cells [14], or to “normal” brain cells undergoing a tumorigenic cascade cross-talking with and recruited by tumor cells [15] is a matter of debate.

Regardless of its origins, it is now accepted that the so-called peritumoral brain zone (PBZ) contributes to GBM heterogeneity and recurrence. Since the PBZ is what actually remains after surgical resection, being the real target of adjuvant therapies, it is evident that a deeper knowledge of its radiological, cellular, and molecular characteristics could be of paramount importance in GBM treatment.

Several strands of evidence have recently shown that the tumor microenvironment plays a crucial role in GBM progression and treatment resistance by promoting tumor growth and invasion while suppressing the immune response [16,17]. The immune microenvironment of GBM also contributes to treatment resistance by inhibiting the anti-tumor immune response [17]. Therefore, active research on therapies targeting the tumor microenvironment to enhance the immune response, disrupt tumor–stroma interactions, and make tumor cells more susceptible to treatment is ongoing [18,19,20].

The scope of this narrative review is to summarize the main findings on different aspects of the PBZ microenvironment in glioblastoma.

It is important to underline that, as mentioned above, grade 4 astrocytoma was introduced only in 2021 by the WHO CNS tumor classification [2]. This entity was defined as IDH-mutated glioblastoma in the 2016 WHO classification [21], and before 2016 it was known as “secondary” glioblastoma, that is, the result of an anaplastic transformation and malignant progression of a lower-grade glioma [22,23]. Therefore, the bulk of the literature currently available on the PBZ does not differentiate between glioblastoma and grade 4 astrocytoma.

However, it is a common experience that IDH-wildtype glioblastoma is by far the most common grade 4 glioma (90–95% of grade 4 gliomas); therefore, most of the data included in this review refer to the most common clinical entity.

Ethical Statement: The data included in Figure 1 and Figure 2 were part of a prospective GBM biobanking study authorized by the Chieti-Pescara Local Ethics Committee (EC number: 08/21.05.2020). The patients gave written consent to the use of their radiological and surgical data in anonymous form.

## 2. Definition of the Peritumoral Brain Zone (PBZ)

When dealing with the PBZ (sometimes defined as the peripheral brain zone or as brain adjacent to tumor (BAT)), the first, difficult issue is its definition. The most-used definition of the PBZ is the area around the contrast-enhancing tumor seen in T1-weighted MR scans. This is quite a vague definition, since contrast enhancement is related to blood–brain barrier disruption and is not a fully reliable picture of tumor extension. Moreover, it is common to see a halo of hyperintensity in T2-weighted and FLAIR sequences around the enhancing tumor [24]. This signal abnormality is related to a mixture of increased cellularity/cellular infiltration and vasogenic edema, with tumor cell infiltration and even frank tumor tissue documented in up to 76% and 9% of samples, respectively, in a recent study [25]. On this basis, the surgical concept of “FLAIRectomy” has recently emerged [26,27]. More advanced radiological techniques based on diffusion-weighted sequences (ADC, FA, and DTI) appear to be more sensitive to changes in the peritumoral area and could be correlated with the degree of glioma infiltration [28,29]. The novel field of radiomics, which aims to extract large numbers of quantitative features from medical images using data-characterization algorithms, could help in the pre-operative analysis of the PBZ, and specific PBZ features could be predictive of survival [24,25,26,30,31,32].

In our neurosurgical practice, we defined, and whenever possible systematically sampled, preferentially via en-bloc supratotal resection, the PBZ as apparently normal brain tissue 1–2 cm around glioblastoma margins, defined through surgical magnification and with neuronavigation [14]. Believing in a small, but still significant, survival advantage of supratotal resection also in GBM [33], when regarded as functionally safe (e.g., in right fronto-polar GBMs), we extended our resection up to 3.5 cm from the enhancing tumor margin: tissue analysis from areas more distant from the tumor showed the same alterations found in areas located within 1 cm from the margin of the tumor [34]. Figure 1 shows an exemplificative case with our usual workflow in PBZ sampling.

## 3. Cellular Characteristics of the PBZ

### 3.1. Tumor Cells

An important concept is that the PBZ is not always infiltrated by frank tumor cells at pathological examination. However, even in these cases, cellular and molecular differences from healthy brain tissue can be detected.

GBM cell infiltration of the PBZ is generally seen in 35–45% of cases [15,24], with some authors reporting an even larger percentage of infiltration in their experience [25]. If present, the number of tumor cells detected in the PBZ is generally between 10 and 30% of the total number of cells in the specimen [35].

At surgical resection, tumor cells from the PBZ do not usually fluoresce when 5-aminolevulinic acid (5-ALA) is used [36]. These cells differ phenotypically from those obtained from the core of the tumor [37,38,39]. In particular, neoplastic cells from the PBZ show a lower stem-cell molecular signature than cells from the core of the tumor, differing in expression of CD133, Sox2, nestin, and musashi 1 [37,38,39]. Therefore, in vitro stemness characteristics, such as self-renewal, are reduced in peripheral tumor cells. However, conflicting results have been published about the in vivo tumorigenic potential of tumor cells from the PBZ. If cancer stem cells (CSCs) derived from the periphery of the tumor have shown little if no tumorigenic potential in animal models [40], a population of non-self-renewing tumorigenic cells has been demonstrated from the same area [39]. Tumor cells from the core and the periphery of a GBM also differ in their response to drugs and irradiation in vitro [38,41].

Nestin expression sharply decreases in peritumoral tissue compared to proper tumor areas. Despite being classically considered as a marker of neural stem cells, a mild nestin increase in apparently normal cells of the PBZ was shown [34]. This could be related to reactive changes in normal astrocytic cells, thus indicating an induced “pre-malignant” state. Expression of nestin and CD105 in endothelial cells of neo-formed microvessels of the PBZ morphologically quite similar to those present in the tumor [42] was also demonstrated.

Bastola and coworkers described uniquely elevated markers attributable to GBM core cells (CD44, MYC, HIF-1α, VIM, ANXA1, CDK6, and JAG1) and GBM edge cells (OLIG1, TC2, SRRM2, ERBB3, PHGDH, and RAP1GAP), showing that intercellular cross-talk from core GBM cells promotes aggressiveness of the edge counterparts, with HDAC1 as the initiator of the cross-talk and sCD109 acting as the mediator via HDAC1-C/EBPβ regulation [41].

Other genetic differences between tumor cells in the PBZ and tumor cells from the core of a GBM involve genes related to invasion (Galectin-1, Rac1, Rac3, RhoA GTPases, p27, and avb3 integrin), cell adhesion (CDH20 and PCDH19), migration (SNAI2, NANOG, USP6, and DISC1), and immune or inflammatory responses (TLR4) [43].

Among others, serine protease inhibitor clade A, member 3 (SERPINA3) protein is more expressed in the PBZ, and its expression correlates with poor prognosis. The in vitro knockdown of SERPINA decreases tumor cell proliferation, invasion, migration, transition to mesenchymal phenotype, stemness, and radioresistance [44].

### 3.2. Non-Neoplastic Cells

The majority of cells detected in the PBZ are peculiar non-neoplastic cells that can be found also when there is no evidence of tumor cell invasion [45]. Among these PBZ non-neoplastic cells, we will discuss the characteristics of angiogenesis-related endothelial cells and of the already mentioned reactive astrocytes, microglia, and other inflammatory and stromal cells. These cells are also detected in the core of the tumor and their vital role in controlling the course of pathology has been revealed in the last decade [46].

#### 3.2.1. Endothelial Cells

Angiogenesis is a key feature of glioblastoma and several trials with anti-angiogenic drugs have been conducted in the last years, with limited impact in clinical practice [47]. However, this exuberant new vascularization is very inefficient, leading to highly hypoxic areas within the tumor and associated necrosis, with hypoxia-inducible factor-1 (HIF-1), the main transcription factor activated by hypoxia, upregulating several genes involved in angiogenesis, metabolic reprogramming, cell invasion, immunosuppression, and cancer stem-cell phenotypes [48]. Hypoxia has, therefore, a major role in GBM maintenance and progression [49].

Neo-angiogenic potential is shared by GBM cancer stem cells derived from both the core and the PBZ [50]. Moreover, neo-angiogenesis occurs in the peritumoral tissue even in the absence of cells with neoplastic morphology. This is suggested by the expression of HIF-1α, HIF2α, vascular endothelial growth factor (VEGF), and VEGF receptors (VEGFR1 and VEGFR2) in both GBM and peritumoral tissue, indicating that both areas contain, to some extent, cells that are either responsive to angiogenic stimuli or able to trigger angiogenic response. However, the vascular characteristics of the PBZ differ from those of the tumor core. Although nestin, CD105, and CD34 are clearly expressed in endothelial cells in the PBZ, HIF-1α and VEGFR-1/2 expression is either low or negative [51]. Also, VEGF-A expression in the PBZ is low compared with that in the core. VEGF-A expression is related to HIF-1α, and this axis plays a pivotal role in the development of the immunosuppressive microenvironment by inhibition of the maturation of dendritic cells, stimulation of the proliferation of regulatory T cells, and suppression of anti-tumor immunity [52,53].

#### 3.2.2. Reactive Astrocytes

Reactive astrocytes are astrocytes that undergo morphological, molecular, and functional changes in response to pathological situations in surrounding tissue [54]. Useful markers for reactive astrocyte detection are GFAP, CD274, proliferation markers (such as Ki67, PCNA, and BrdU incorporation), aldehyde dehydrogenase-1 L1 (ALDH1L1), glutamine synthetase (GS), and aldolase-C (ALDOC) [54,55]. In gliomas, astrocytes display a reactive phenotype due to activated microglia induction and glioma cell contact, working synergistically to promote GBM proliferation and invasiveness [56,57]. Reactive astrocytes are also believed to offer chemoprotection to glioma cells via gap-junction communication, and especially connexin 43 [58,59], and immunoprotection via paracrine secretion of several factors, including tenascin-C [60] and IL-10 [61]. A higher expression of factors related to reactive astrocytes is found in the PBZs of patients with shorter survival after surgical resection compared to longer-term-survival patients [62].

#### 3.2.3. Glioma-Associated Microglia/Macrophages (GAMs)

Microglia, which are the brain-resident immune cells, and peripheral blood macrophages are the most common non-neoplastic cells detected in the GBM microenvironment [63]. As per reactive astrocytes, the pro-tumor role of microglia and macrophages in GBM growth and progression has been recently postulated [64]. In fact, these cells could contribute to immune evasion, growth, and invasion of GBM [65,66].

Three possible phenotypes have been described for microglia/macrophages [67,68,69]: the M0 phenotype, or resting microglia, contributing to the maintenance of a healthy environment for neuronal function; the activated M1 phenotype, or pro-inflammatory microglia, characterized by the ability to release pro-inflammatory cytokines/mediators, such as IL-1β, IL-6, TNF-α, CCL2, reactive oxygen species (ROS), and nitric oxide (NO); and the activated M2 phenotype, or anti-inflammatory microglia, associated with the ability to produce anti-inflammatory and immune-suppressive factors, including ARG-1, Ym1, and CD36, as well as upregulate the cell surface markers CD163, CD204, and CD206 and anti-inflammatory cytokines, such as IL-10. M2 GAMs also secrete transforming growth factor beta (TGF-β), VEGF, and matrix metalloproteinase 9 (MMP9).

However, a rigid, dichotomic characterization (M1 vs. M2 phenotypes) is considered an oversimplification, and a new view based on the coexistence of multiple states is currently encouraged by experts in the field [70]. Glioma and microglia interaction result in a dynamic stimulus-dependent microglia phenotype expression [71,72,73]. In a condition mimicking the late stage of GBM in vitro, microglia present as a mixture of polarization phenotypes (M1 and M2a/b), while microglia exposed to factors resembling the early stage of pathology show a more specific pattern of activation, with increased M2b polarization status and upregulation of IL-10 only [71].

Ex vivo, glioma-associated microglia/macrophages (GAMs) show a heterogeneous M1/M2 polarization within the tumor and the PBZ. CD163+, M2-polarized GAMs, are highly expressed in the tumor core, with a core-to-PBZ decreasing gradient [74,75]. A correlation between high CD163 expression and reduced overall survival has been shown in GBM and grade 3 gliomas but not in grade 2 gliomas [74,76]. Nonetheless, iNOS + GAMs, that is, M1-polarized GAMs, also show a higher concentration in the tumor core compared with the PBZ [74], confirming that these cells are recruited by the tumor and that the notion of a stimulus-dependent microglia phenotype should replace the oversimplified M1/M2 polarization paradigm [72,73].

Regarding the role of GAMs in GBM progression, intense GAM–glioma cross-talk has been shown [77,78]. Most GAMs arise from circulating monocytes, whose recruitment and differentiation into GAMs is supported by CCL2 and CSF-1. Indeed, glioma cells release several other chemoattractive factors (e.g., MCP-1 and 3, HGF/SF, CX3CL1, GDNF, ATP, SDF-1, GM-CSF, etc.) to recruit GAMs to the tumor tissue. Moreover, tumor cells secrete inflammatory factors, chemokines, and signal molecules to promote the M2 polarization of GAMs, such as IL-4, CCL2, and TGF-β [79]. In turn, GAMs release several other factors to promote glioma cell invasion, such as TGF-β, STI1, EGF, IL-6, and IL-1β. In particular, TGF-β triggers the release of pro-MMP2 and versican from glioma cells and a cascade that ultimately leads GAM cells to produce active MMP2 and MMP9, which enhance degradation of the extracellular matrix and glioma invasion [77].

#### 3.2.4. Tumor-Infiltrating Lymphocytes (TILs)

Other inflammatory cells found in the GBM microenvironment are T-cells and NK cells [80]. However, GBM is often described as a “cold tumor”, due to the prevalence among tumor-infiltrating lymphocytes (TILs) of regulatory T-cells, a specialized subpopulation of T cells that act to suppress the immune response, thereby maintaining homeostasis and self-tolerance, and a relative lack of cytotoxic T-cells (CD8+). Moreover, it has been shown that TILs in GBM are more immunosuppressed than their peripheral counterparts, and dysfunction of these TILs, which often show an “exhausted” phenotype, is a hallmark of GBM [81].

T-cells, identified through CD3 positivity, are more prevalent in the tumor core than in the PBZ, with a similar gradient to CD163+, M2-like GAMs [75]. These M2-like immunosuppressive GAMs, as well as reactive astrocytes, release several immunosuppressive factors, such as IL-10 and TGF-β, both of which inhibit cytotoxic T-cell functions [82,83]; arginase, which inhibits T-cells through arginine depletion from the tumor microenvironment [84]; and indolamine 2,3-dioxygenase (IDO), which acts to recruit regulatory T-cells and inhibit cytotoxic T-cells through tryptophan depletion [85,86]. Our research group recently showed that IDO is expressed almost exclusively in the tumor core, with substantially no staining in the PBZ, and that IDO positively localizes with CD163+ cell clusters, confirming GAMs as the source of IDO secretion [75]. Also, a high HIF-1α concentration in the tumor core attracts regulatory T-cells and promotes the migration and differentiation of immunosuppressive GAMs, resulting in further suppression of cytotoxic T-cells.

In summary, when compared with the tumor core, the PBZ shows a higher density of CD8+ cytotoxic T-cells, a lower density of Foxp3+ regulatory T-cells, and a similar number of CD4+ T-cells, which include both regulatory and helper T-cells [87]. Interestingly, cytotoxic T-cells show a significantly lower expression of programmed cell death-1 (PD-1) in the PBZ compared with the tumor core [87]. Indeed, expression of PD-1, a T-cell surface protein serving as an immune checkpoint which downregulates the immune system to prevent auto-immunity [88,89], is upregulated by VEGF-A [90]. Of note, hypoxia in the tumor core induces VEGF expression via the HIF-1α pathway, which also induces the expression of the PD-1 ligand (PD-L1) [91].

#### 3.2.5. Glioma-Associated Stromal Cells (GASCs)

It has been shown that in systemic carcinomas a significant component of the tumor stroma is constituted by cancer-associated fibroblasts (CAFs), also known as myofibroblasts [92]. Recently, GBM-associated stromal cells (GASCs) from histologically tumor-free PBZs with phenotypic and functional properties similar to CAFs have been isolated [93]. The nomenclature of these cells is still debated, as several names, including glioma-associated human MSCs (GA-hMSCs), glioma-associated or glioblastoma-derived MSCs (gbMSCs), glioma stroma MSCs (GS-MSCs), brain tumor-derived MSCs (BT-MSCs), mesenchymal stem-like cells (MSLCs), tumor MSCLs (tMSLCs), glioma stromal MSLCs (GS-MSLCs), CAF-like cells, and glioblastoma-associated stromal or glioma-associated stem cells (GASCs), have been used in the literature [94], inducing confusion and possible errors. Probably, the most problematic name used by some authors is glioma-associated stem cells, which could erroneously be interpreted as a synonym of glioma stem cells (GSCs). However, GASCs and GSCs are genetically different, as GASCs are typically diploid and do not harbor the genetic alterations commonly seen in GSCs, such as loss of chromosome 10 or gain of chromosome 7 [95,96].

Interest in GASCs is related to the evidence that their probable progenitor cells, namely, mesenchymal stem cells (MSC), also known as bone marrow-derived mesenchymal stem cells (bm-MSC), show anti-tumor behavior [97] and are regarded as carriers for cancer therapy [98]. Nonetheless, there is still an ongoing debate on the progenitor cells of GASCs, which goes beyond the scope of this manuscript, and even their role in GBM behavior is debated [95,99,100,101].

However, several experimental data seem to confirm that tumor-resident GASCs have a pro-tumorigenic role. Indeed, GASCs have a complex cross-talk with GBM tumor cells [102]. Tumor cells recruit GASCs and enhance their angiogenic potential via platelet-derived growth factor-BB (PDGF-BB), Matrix Metalloproteinase 1 (MMP1), IL-8, VEGF-A, and TGF-β1 secretion [103,104,105,106]. In turn, GASCs can facilitate tumor cell migration by remodeling the extracellular matrix [107], drive phenotypic changes in GBM cells, and stimulate GBM cells, leading to greater tumor infiltration [99,108]. The interaction of GASCs with GSCs could even produce fusion cells that were more angiogenic than the parental cells in animal models [109].

Two subtypes of GASCs differentiated by DNA methylation upon transcriptome analysis can be isolated: GASCs-A and GASCs-B [37]. The latter show pro-carcinogenic properties both in vitro and in vivo, promoting the development of tumor cells and endothelia. It has been shown that GASCs can stimulate glioma malignancy also through the M2 GAMs and are associated with the level of immune checkpoints in the glioma microenvironment [110].

### 3.3. Summary of Cellular Characteristics of the PBZ

Several cellular types play a role in GBM development and behavior. The PBZ shares most of these players with the tumor core. However, most of the involved cell types show different characteristics depending on their spatial distribution. Overall, an increasing malignant-behavior gradient is present from the PBZ to the tumor core, where GBM expresses the apex of its immunosuppressive properties. Hypoxia has a key role, orchestrating the molecular landscape of the tumor core. In turn, the tumor core influences angiogenesis in the PBZ, which still has some preserved immune properties that are progressively lost when approaching the tumor core.

Figure 2 shows a second exemplificative clinical case of a GBM undergoing supratotal resection and sampling of the PBZ, with a graphical rendering of the main cellular populations in the tumor core and the PBZ.

Table 1 and Figure 3 summarize the main cellular properties of the peritumoral brain zone in glioblastoma and the cross-talk between different cellular populations to promote tumor growth, invasion, and immune-, chemo-, and radioresistance.

## 4. Molecular Characteristics of the PBZ

Here, we report the most relevant findings from immunohistochemistry analysis and the main genomic and transcriptomic characteristics of the PBZ in GBM. Some of these data have already been mentioned in the previous subsections as part of the narrative.

### 4.1. Immunohistochemistry

Glial fibrillary acidic protein (GFAP) immunolabeling in the PBZ is common and associated with reactive astrocytes.

Ki-67 immunoreactivity is dependent on the presence or not of infiltrating tumor cells.

As already mentioned in the previous section, neo-vascularization markers, such as nestin and CD105, are expressed in the PBZ regardless of the presence of tumor cells [42]. Indeed, nestin is expressed also in reactive astrocytes and normal glia of the PBZ [34]. The presence and activation of mitogen-activated protein kinases both in tumors and the PBZ was also shown. These are signal transduction pathways activated by growth factors. In particular, the presence and activation of the extracellular signal-regulated kinase ERK1/2 and stress-activated protein kinases/c-Jun NH2-terminal kinases (JNKs) were detected [34,111]. As per nestin, this was shown independently of the presence of neoplastic cells and not only in reactive astrocytes, but also in apparently normal glial cells. ERKs play a crucial role in transducing growth factor signals to the nucleus and are involved in a wide range of biological responses, including cell proliferation, differentiation, and motility; thus, this pathway might be involved in tumor progression. The role of JNK is more controversial. It is strongly expressed in the tumor core. Enhanced JNK activation has been found in brain tumor cell lines in response to epidermal growth factor, and the isoform JNK2a2 promotes phenotypes associated with tumorigenesis and proliferation [112,113]. However, JNK is also associated with apoptosis [114]. Prolonged OS has been shown in GBM patients who presented in their PBZs a higher expression of JNK in its activated form (pJNK), a higher ratio between pJNK and total JNK expression (pJNK/tJNK), and higher pJNK/nestin and (pJNK/tJNK)/nestin ratios [34].

Increased expression of HIF-1α as well as other hypoxia-regulated proteins, such as VEGF, in the PBZ have been correlated with reduced progression-free survival after surgery and adjuvant treatment [115].

The tumor border, sometimes called the infiltration zone, showed strong staining for the “signal transducer and activator of transcription” STAT-1, both in tumor elements and in reactive astrocytes and microglia of the non-invaded brain [116]. STAT proteins are intracellular transcription factors activated by the membrane cytokine receptor JAK (the JAJK/STAT pathway) that mediate many aspects of cellular immunity, proliferation, apoptosis, and differentiation. STAT-1 can be induced by interferon γ and growth factors, such as the interleukins IL-6 and IL-10, growth hormone, thrombopoietin, and insulin-like growth-factor binding proteins (IGFBP-3). The role of STAT-1 expression in GBM and the PBZ is controversial. Its involvement in apoptosis could constitute a neuroprotective mechanism and could favor response to chemotherapy [116]. However, a correlation between increased STAT-1 expression and shorter overall survival has been documented [117]. This could be related to overexpression of STAT-1 in its inactivated form, as suggested by the restriction of the staining to the cytoplasm of tumor cells, indicating that in glioblastomas STAT-1 is not translocated to the nucleus [116].

Microglia of the PBZ also avidly stain for A1 adenosine receptor (A1AR), one of the receptors for the regulatory nucleoside adenosine expressed in response to cellular stress and damage, such as episodes of tissue hypoxia and inflammation. A1AR has anti-inflammatory properties and its expression seems to play an anti-tumorigenic role mediated by microglia cells [118].

Our research group recently showed that galectin-3-binding protein (LGALS3BP), a secreted, hyperglycosylated protein, is highly expressed, with a cytoplasmatic pattern, in the GBM tumor core, while it is barely detectable in the PBZ, where it is primarily confined to perineuronal glial cells [119]. This secreted protein has several proposed roles in contributing to a pro-tumorigenic microenvironment (for a review, consult Capone et al. [120]). While we could not show a correlation between this expression and any significant clinic-pathological finding, we showed that tumor cells are the source of extracellular vesicle (EV)-associated LGALS3BP. Indeed, increased levels of EV-associated LGALS3BP were found in glioma patients compared with healthy donors, with increasing concentrations associated with increasing glioma grade [119,121]. These findings are relevant, since they show that GBM cells secrete LGALS3BP, probably to favor tumor progression in the PBZ, and circulating EV-associated LGALS3BP could have a role in “liquid biopsy” from plasma sampling in GBM diagnosis and monitoring. Moreover, LGALS3BP could be a target for non-internalizing antibody–drug conjugates [122], which are considered as a promising new frontier of personalized cancer medicine [123].

### 4.2. Genomic and Transcriptomic Characteristics

Studies from the French Grand Ouest Glioma Project could not find significant genomic alterations in non-infiltrated PBZs [24,35], while tumor-cell-infiltrated PBZs showed some genomic alterations in common with their tumor cores, in particular, chromosome 7 polysomy, EGFR amplification, and chromosome 10 deletion.

However, other research groups, including ours, could detect some abnormal gene expression in PBZs not invaded by tumor cells upon pathological analysis [15,124]. Indeed, when matching genomic alterations of a tumor core with the corresponding PBZ in a single patient, a variable amount of correlation was found [124,125]. Non-infiltrated PBZs and enhancing tumors were shown to share some genomic anomalies: del(1p36), del(2p21), MDM2 and CDK4 amplification, and amplification of 15q24.1, 219, and 222 [14]. CDK4 is a known cancer promoter gene involved in several systemic cancers, including breast carcinoma, that participates in the Cyclin D1 (CCND1)–CDK4/6–CDKN2A (p16INK4A)–Rb axis, which is often altered in the GBM [126]. Overexpression of CDK4 in the GBM PBZ is related to poor outcomes [125].

Indeed, gene expression shows only a partial overlap between the tumor core and the PBZ. Both tissues show upregulation of genes related to angiogenesis, the extracellular matrix, and cellular senescence and with stemness features, while downregulation is seen in tumor-suppressor genes, such as SLC17A7 and CHD5 [127]. Moreover, the PBZ shows upregulation of UBE2C, NUSAP1, IGFBP2, SERPINA3, and PBK genes when compared to the tumor core [127].

When summarizing transcriptomic data, it should be kept in mind that the possible infiltration of tumor-infiltrating cells or even of glioma stem cells could influence the reported data, as only a few manuscripts have tried to minimize this possible bias. We will therefore now focus on these studies.

When comparing tumor cores, non-invaded PBZs, and normal white matter from non-glioma patients, several differences in gene expression were found [15]. Briefly, tumor and PBZ samples showed significant changes in the expression of 1323 genes. Among genes with at least a 10-fold difference in expression level, 20 genes were overexpressed in GBM when compared to the PBZ, while 45 genes were downregulated. High overexpression was seen for genes involved in angiogenesis (VEGF and ANGPT2), genes associated with cell growth (IGFBP2 and GAP43) and with the cell-cycle activator CKS2, and genes encoding proteins associated with extracellular matrix formation, including COL4A1, COL4A2, COL1A1, COL3A1, and COL1A2. The majority of the 45 genes highly downregulated in GBM compared to the PBZ were involved in the development of the nervous system (MOG, RAPGEF5, GRM3, SH3GL3, NINJ2, UGT8, MOBP, and MBP).

PBZs with no tumor cell infiltration showed significant differences in the expression of 57 genes compared to white matter from control patients: 15 genes were overexpressed and 42 genes were downregulated in the PBZs. EGFR expression levels showed the largest difference between healthy white matter and PBZs, being highly expressed in the latter. Moreover, genes belonging to two main relevant biological processes were particularly deregulated in the PBZs. In detail, genes associated with growth and proliferation (CSRP2, TAZ, ID3, and DTNA) and cell motility/adhesion (HIST2H2AA, EGFR, IGFBP5, VCAM1, and CD99) were upregulated, while genes involved in neurogenesis (SYNJ1, NBEA, SERPINI1, CNTNAP2, and RELN) and several tumor-suppressor genes (BAI3, PEG3, PRDM2, and RB1CC1), along with the natural killer receptor KLRC1, were downregulated in the PBZs.

Interpretation of the above data is difficult, with the postulate of a “pre-cancerous” state of brain tissue surrounding the tumor under the influence of cross-talk between tumor cells and normal cells resulting in a recruitment of the latter cells in GBM proliferation and growth being an intriguing possibility.

In another work, GBM de novo tumor cores and PBZ transcriptomes were compared through a serial analysis of gene expression (SAGE) technique, and their miRNomes were compared through a subsequent microRNA deep sequencing [62]. The results were analyzed separately for short- (<36 months) and long-term (≥36 months) survivors. In agreement with other groups’ findings [35,45], the transcriptional features of the PBZ showed a predominance of the TCGA neural subtype [128], regardless of the presence or not of tumor cell infiltration and with no correlation with the tumor core features. The tumor center showed overexpression of several molecules belonging to the “mesenchymal signature” of glioblastoma, likely due to microglia and reactive astrocytes infiltration.

Chemokine (C-X-C motif) ligand 14 (CXCL14) RNA was found to be overexpressed in the tumor cores and PBZs of both long- and short-term survivors when compared to healthy white matter from non-glioma patients [62]. However, while in long-term survivors its expression resulted in comparable tumor centers and PBZs, in short-term survivors it was overexpressed in the majority of tumor center samples compared to paired PBZs [62]. CXCL14 is a small cytokine, mainly contributing to the regulation of immune cell migration [129], whose transcription is induced in microglia activated towards the pro-invasive, immunosuppressive (M2) phenotype by glioma cells in experimental conditions [130].

Transforming growth factor beta-induced (TGFBI) protein is considered a feature of the mesenchymal GBM subtype. TGFBI RNA was found to be overexpressed in both the tumor cores and PBZs of all patients compared to healthy white matter from controls, with significant overexpression in the tumor cores between long-term and short-term survivors [62]. TGFBI is a protein secreted in the extracellular matrix, the mediator of the non-SMAD-mediated TGFβ signaling pathway in GBM [131], that interacts with several other extracellular matrix components, such as collagens, in cell–cell or cell–substrate adhesion. TGFBI is preferentially secreted by M2-like GAMs and indicates poor prognosis in GBM patients [132].

Giambra and colleagues found some peculiar gains in some loci in the short arms of chromosomes 11 and 16 in PBZs, not shared by tumor cores, focusing on the possible role of the EXT2 gene, which is involved in the biosynthesis of the heparan sulfates—glycosaminoglycans distributed on the cell surfaces and in the extracellular matrices of most tissues which could be involved in angiogenesis and tumor proliferation [125].

mRNA analysis of tumor cores compared with their own PBZs showed several signs of a higher participation of reactive cell types, such as microglia and reactive astrocytes, which could be depicted as a generally more “mesenchymal” feature, in short-term survivors but not in long-term survivors [62]. Comparing the PBZs of short-term and long-term survivors, indications of a possible higher contribution of tumor “stromal” cells, such as IGFBP5 mRNA expressed by reactive astrocytes, in short-term cases were documented. In addition, a sound overexpression of the extracellular protein lumican in long-term survivors’ PBZs as compared to short-term survivors’ ones was detected. This stromal protein has recently been shown to positively correlate with prolonged survival after tumor resection in pancreatic ductal adenocarcinomas due to its limiting role in EGFR-expressing pancreatic cancer progression [133].

MicroRNA analysis showed overexpression of miRNAs involved in TGFβ active pathways and mediators of a general state of immune escape typically sustaining glioma growth in all tumor core and PBZ samples compared with heathy white matter. In particular, overexpression of miR-106b and miR-93, found to target a ligand of the activating receptor of natural killer (NK) cells (NKG2DL), and of the TGFβ-induced miR-183, reported to suppress tumor-associated NK cells, was detected. The latter “oncomiR” (miR-183-5p), together with the chromosome 7-located miR-182-5p and miR-96-5p, showed differential expression between PBZ samples with detectable tumor cell infiltration and non-infiltrated samples. These three miRNAs are recognized as mediators of TGFβ signaling in GBM [134].

Underexpression of miR-128a, miR-181a, miR-181b, and miR-181c in tumor centers compared to PBZs was shown [135]. These miRNAs show peak expression at the adult stage when compared to embryonic and early post-natal stages: this might indicate some non-random correlation with a de-differentiated state of tumor cells (having lost their “adult brain signatures”).

Piwecka and collaborators showed that miR-625 was downregulated in the PBZ compared to normal tissue but not in the tumor core [136]. MiR-625 overexpression suppresses cell proliferation and colony formation, induces G0/G1 arrest, increases chemosensitivity to temozolomide by targeting AKT2 in glioma cell lines, and suppresses tumor growth and angiogenesis in animal models [137].

The target-mimetic, sponge/decoy function of long non-coding RNAs (lncRNAs) on microRNAs was recently uncovered: they can act as miRNA sponges, reducing their regulatory effect on mRNAs [138]. A higher expression of the highly expressed long non-coding RNA in glioblastoma (lncHERG), acting as an miR-940 sponge, was found to indicate a lower survival rate and poorer prognosis, being related to cell proliferation, migration, and invasion, both in vitro and in vivo. LncHERG expression was shown to be higher in glioblastoma than non-tumor tissues, while the tumor suppressor miR-940 was, on the contrary, shown to be downregulated in glioblastoma tissues compared to peritumoral tissues [139].

Very few data on PBZ proteomics are available [140], with some proteins being overexpressed in the PBZ, in particular, the histone H3F3A and the crystallin B α-chain (CRYAB), both recognized as oncogenes.

The main immunohistochemistry, genomic, and transcriptomic characteristics of the PBZ are summarized in Figure 4.

## 5. Discussion

Analysis of the cellular and molecular characteristics of the PBZ is fundamental not only for understanding GBM pathophysiology but also to enhance therapeutic options, since glioblastoma will almost inevitably recur from this tissue after extensive surgical resection and radio-chemotherapy. Indeed, several molecular mechanisms involved in the tumor core–PBZ cross-talk are currently regarded as potential therapeutic targets.

The key concepts of this review are:-The PBZ is the real target of post-surgical therapies; therefore, understanding it is of paramount translational significance.-Only in about one third of cases is it possible to detect tumor cells via microscopic analysis of the PBZ; nonetheless, we can observe distinct abnormal features also in cases of no infiltration.-If some data on the PBZ can suggest the influence on the features of this area of tumor cells migrated well beyond the tumor border, diluted in brain parenchyma, and possibly having a “dormant” feature and therefore liable to be missed by pathological examination, others suggest the recruitment of surrounding cells reprogrammed towards a neoplastic phenotype.-Angiogenesis and hypoxia play a pivotal role in influencing the tumor and PBZ microenvironment.-An increasing immunocompromised gradient is seen from the PBZ to the tumor core.-The role of inflammatory cells has been illuminated in recent years, and several alterations in the PBZ can be related to an activation of microglia and astrocytes that could promote tumor growth and invasion.-The transcriptomic profile of the PBZ is extremely complex, since this tissue differs from normal white matter, shares some tumor core characteristics, and possibly shows some peculiar alterations. Overall, the PBZ seems to be “half-way” on the road towards malignancy. However, the data are highly heterogeneous due to possible infiltration of glioma cells/glioma stem cells. Single-cell analysis could improve our understanding.

In summary, current data show that GBM actively induces transformation of the surrounding environment, inducing degradation of the extracellular matrix and recruiting astrocytes, microglia, macrophages, and regulatory T cells to promote an immunosuppressive environment. However, these mechanisms have a decreasing gradient at increasing distance from the tumor necrotic core, with the presence of still-competent immune cells in the PBZ, namely, a higher number of PD-1 negative CD8+ cytotoxic T-cells. Currently, several ongoing trials are trying to enhance self-immunity [141,142]. Since the post-operative target of therapies is the PBZ, future research should focus on its molecular mechanisms, which only partially overlap with those of the tumor core.

Indeed, while a huge amount of data are available on the glioblastoma core, the PBZ is still relatively poorly investigated, with increasing interest in it having been shown only in the last few years in the medical literature. Moreover, it is difficult to reproduce the PBZ microenvironment in both in vitro and in vivo animal studies, as patient-derived cell lines can usually be obtained only from the most aggressive, tumor core-like GBM cells that infiltrate the PBZ. Therefore, most of the data on the PBZ can be derived by histopathologic and multi-omics analyses of appropriately selected surgical samples. Hence, it is fundamental for neurosurgeons to be directly involved in neuro-oncology research in order to opportunely select patients from whom PBZs can be safely sampled, pre-plan PBZ tissue resection with an adequate intraoperative methodology, and store the samples in an appropriate form for the planned biological analyses.

## 6. Conclusions

A better understanding of the PBZ would be of high translational potential. Indeed, the pathophysiology of the PBZ can be indicate early events of GBM tumorigenesis and help in detecting its initial alterations. From a surgical point of view, understanding of PBZ characteristics could help in the development of intraoperative techniques aiming to detect areas of alteration surrounding the macroscopically evident tumor. From a translational and therapeutic point of view, new molecular targets and possibly personalized adjuvant therapies could be developed with a better understanding of the PBZ, which is the real target of post-surgical therapies.

## Figures and Tables

**Figure 1 cancers-15-05460-f001:**
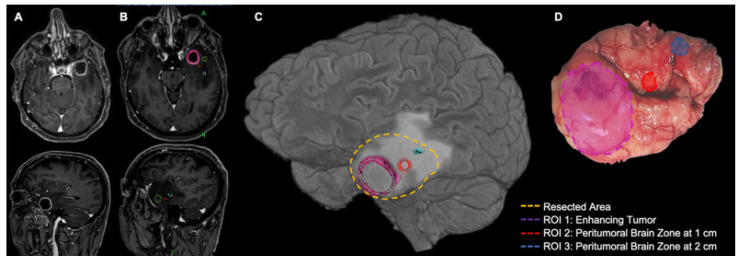
Exemplificative case of peritumoral brain zone (PBZ) sampling in left temporal pole glioblastoma. (**A**) Pre-operative contrast-enhanced T1w MR images. Note the tumor core composed of an inner necrotic area and an enhancing border. Peritumoral brain zone is beyond the enhancing border. (**B**) Pre-operative MR scans were imported into a neuronavigation workstation and several regions of interest (ROIs) were detected: enhancing tumor (pink ROI), two ROIs from PBZs at 1 cm from enhancing tumor (red and green ROIs), and one ROI from PBZ at 2 cm from enhancing tumor (blue ROI). (**C**) Three-dimensional surface rendering of the case with superimposed dotted lines showing the supratotal en-bloc resection (yellow dotted line) that includes enhancing tumor (pink dotted line) and the preselected ROIs of PBZ at 1 cm and 2 cm from enhancing tumor (red and blue dotted lines, respectively). Please note that FLAIR alteration goes far beyond enhancing tumors and that sampled PBZ ROIs are located within FLAIR alterations. A complete “FLAIRectomy” could not be performed in this case due to the risk of postoperative language deficits related to posterior superior temporal gyrus (“Wernicke area”) involvement in FLAIR alteration. (**D**) Surgical sample with ROI overlay.

**Figure 2 cancers-15-05460-f002:**
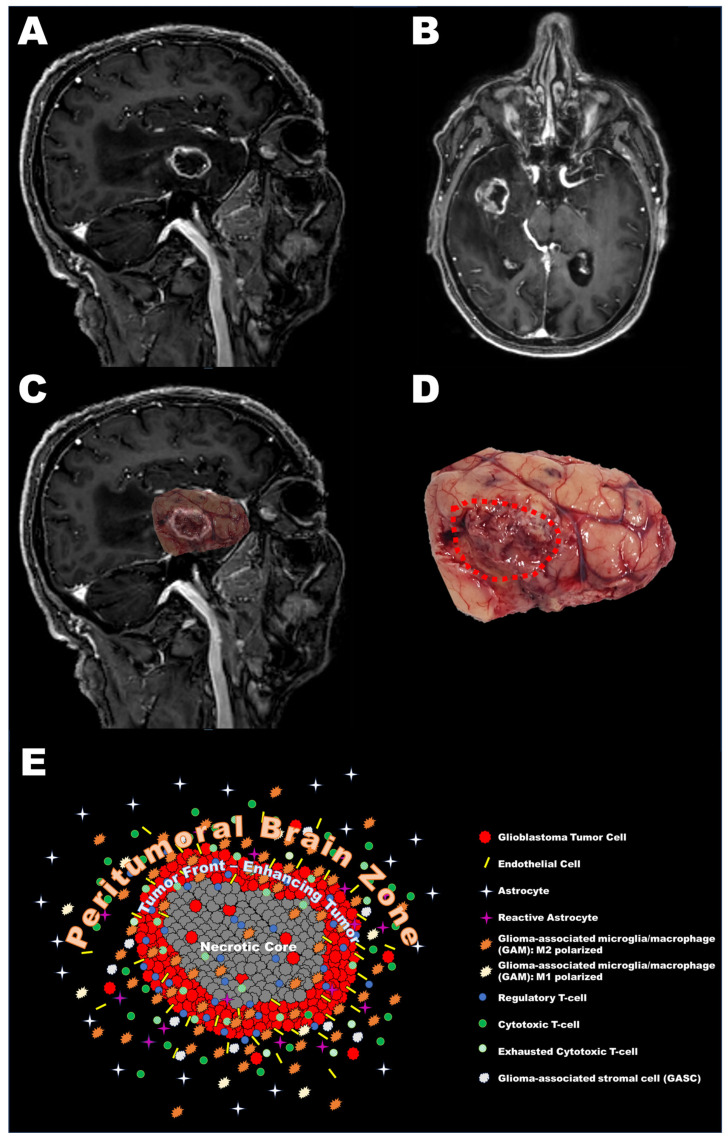
Right temporal glioblastoma. (**A**) T1w gadolinium enhanced MRI, sagittal view. (**B**) Axial plane. Note the necrotic core, the enhancing ring, and the peritumoral signal alteration (hypointense in T1-weighted scans). (**C**) Overlay of surgical sample on pre-operative scan. Please note the correspondence of the tumor core, seen as altered brain parenchyma, even at gross visual inspection, with the enhancing area. (**D**) Surgical sample with an overlayed contouring of tumor core (red dotted line). (**E**) Graphical rendering of the cellular microenvironment seen when analyzing slices from a supratotal GBM resection. Cellular types are detailed on the right side. The distribution of cellular types between tumor core and peritumoral brain zone (PBZ) resembles evidence in the literature as per the main text of this manuscript.

**Figure 3 cancers-15-05460-f003:**
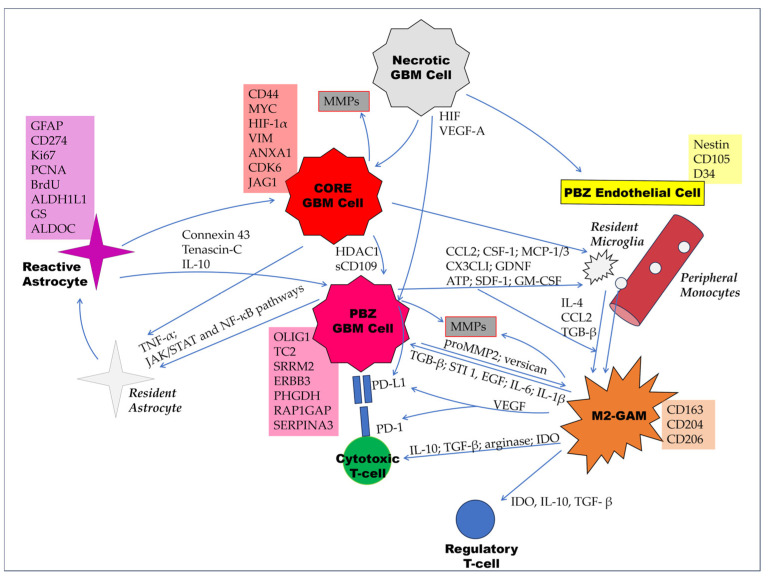
Main underlying mechanisms of the interplay of immune and other cells within the microenvironment of the peritumoral brain zone (PBZ) in GBM. Main cell markers are shown in rectangular boxes at the side of any cell type. Core GBM cells promote aggressiveness of the PBZ counterparts, with sCD109 acting as the mediator of HDAC1-C/EBPβ pathway. GBM cells induce reactive astrocytes via TNF-α secretion and through direct cell–cell contact by a variety of signaling pathways, including the JAK/STAT and NF-κB pathways. In turn, reactive astrocytes offer chemoprotection to glioma cells via gap-junction communication (connexin 43) and immunoprotection via paracrine secretion of tenascin-C and IL-10. The necrotic core induces endothelial cell proliferation via the HIF-VEGF-A axis, which also induces MMP production and PD-L1 expression in GBM cells. Glioma cells attract resident microglia and peripheral monocytes and induce their transformation into M2-polarized glioma-associated microglia/macrophages (GAMs). The cross-talk between M2-GAMs and glioma cells leads to MMPs production and PD-L1 overexpression in GBM cells.M2-GAMs also suppress the immune system by secreting IDO, IL-10, and TGF-β, which activate regulatory T-cells and induce PD-1 expression by cytotoxic T-cells. These mechanisms are similar between the tumor core and the PBZ, with a decreasing gradient of immunosuppressive properties from the center to the periphery.

**Figure 4 cancers-15-05460-f004:**
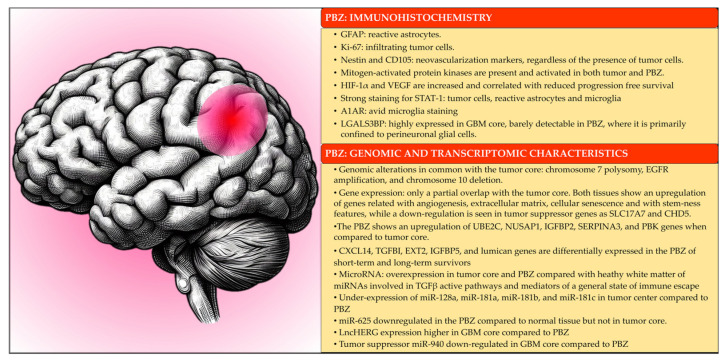
Main immunohistochemistry, genomic, and transcriptomic characteristics of peritumoral brain zone (PBZ).

**Table 1 cancers-15-05460-t001:** Summary of main cellular characteristics of PBZ.

Peritumoral Brain Zone Cellular Characteristics		
Cell Type	Details	Main References
Tumor-infiltrating cells	Seen in 35–45% of cases, representing 10–30% of cells in the specimen No/low fluorescence at 5-ALA Low stem-cell molecular signature Show a different expression of tumor markers compared with tumor core neoplastic cells Different in vitro response to drugs and radiation compared with tumor core neoplastic cells	Mangiola 2013 [15]; Lemée 2015 [24] Idoate 2011 [36] Clavreul 2015 [37]; Piccirillo 2012 [39]; Bastola 2020 [41] Glas 2010 [38]; Nimbalkar 2021 [44]
Endothelial cells	Angiogenesis occurs in both tumor core and PBZ, also with no infiltrating tumor cells HIFs are pivotal in GBM microenvironment As in tumor core, in PBZ, endothelial cells express nestin, CD105, and CD34 Compared to tumor core, in PBZ, endothelial cells express low or null HIF-1α, VEGF-A, and VEGFR-1/2	D’Alessio 2016 [50] Domènech 2021 [48]; Shi 2023 [49] Tamura 2018 [51]
Reactive astrocytes	Promote GBM proliferation and invasiveness Chemoprotection through gap junctions and connexin 43 Immunoprotection through tenascin-C and IL-10 Reduced survival in patients with higher density in PBZ	Placone 2016 [56]; Liddelow 2017 [57] Lin 2016 [58]; Munoz 2014 [59] Huang 2010 [60]; Fujita 2008 [61] Fazi 2015 [62]
Glioma-associated microglia and macrophages	Recruited and polarized by tumor cells Contribute to immune evasion, growth, and invasion of GBM M2-polarized GAMs (anti-inflammatory phenotype) are prevalent, with a decreasing gradient from core to PBZ. They have a prominent pro-tumorigenic role Also, M1-polarized GAMs (pro-inflammatory phenotype) are recruited by GBM, with a decreasing core-to-PBZ gradient, similar to M2 GAMs	Annovazzi 2018 [76] Hambardzumyan 2016 [77] Hussain 2006 [65]; Watters 2005 [66] Ge 2020 [79]; Lisi 2017 [74] Rahimi Koshkaki 2020 [75]
Tumor-infiltrating lymphocytes	PBZ shows higher number of PD-1-negative CD8+ cytotoxic T-cells Cytotoxic T-cells in tumor core have an “exhausted” phenotype and the majority are PD-1+ (immunosuppressive) Tumor core has higher density of regulatory T-cells (immunosuppressive) Overall, more TILs in tumor core with same gradient of GAMs	Tamura 2018 [87] Woroniecka 2018 [81] Rahimi Koshkaki 2020 [75]
Glioma-associated stromal cells	Similar to cancer-associated fibroblasts seen in systemic cancers Several names used in the literature Unknown progenitor cells (mesenchymal stem cells?) GASCs and GSCs are genetically different, as GASCs are typically diploid and do not harbor the genetic alterations commonly seen in GSCs, such as loss of chromosome 10 or gain of chromosome 7 Tumor cells recruit GASCs and induce their angiogenic potential GASCs facilitate tumor cell migration by remodeling the extracellular matrix and inducing phenotypic changes in GBM tumor cells GASCs also interact with M2-polarized GAMs	Clavreul 2012, 2014, 2020 [93,94,95] Hossain 2015 [96] Schichor 2006 [103] Zhang 2021 [106] Birnbaum 2011 [105] Ho 2009 [104] Lim 2018 [107] Cai 2021 [110]

Abbreviations: 5-ALA: 5-aminolevulinic acid; GAMs: glioma-associated microglia and macrophages; GASCs: glioma-associated stromal cells; GBM: glioblastoma; GSCs: glioma stem cells; HIFs: hypoxia-inducible factors; PBZ: peritumoral brain zone; TILs: tumor-infiltrating lymphocytes; VEGF: vascular endothelial growth factor; VEGFR: vascular endothelial growth factor receptor; PD-1: programmed cell death protein 1.

## Data Availability

Data are contained within the article.

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
