# Peer review of "Current Knowledge about the Peritumoral Microenvironment in Glioblastoma"

_cancers, 2023, doi:10.3390/cancers15225460_

Round 1

Reviewer 1 Report

Comments and Suggestions for Authors

Ref: cancers-2654537

The manuscript entitled: “Current knowledge about peritumoral microenvironment in glioblastoma” by Trevisi and Mangiola, is a narrative review article that summarizes the most relevant findings about the biological characteristics of PBZ currently available under the medical perspective. This is an interesting study that fits within the scope of the journal and the special issue. I recommend this manuscript to be accepted for publication in the Cancers Journal, after some major revisions. Please find below the comments-suggestions and revisions that will help the authors improve the current version of this manuscript:

Major/minor comments:

Abstract:

-Results: since this is review article there is no need for this part in the abstract.

-Lines 29-30: ” Understanding of PBZ could be relevant to develop more effective treatments to prevent GBM recurrence”: since you are talking about pre-malignant microenvironment it would be better to rephrase ”to prevent GBM development and recurrence”.

-Background: It would be better to define the glioblastoma according to the new WHO2021 Classification at the beginning by briefly mentioning the different glioma grades in the first paragraph and explaining the GBM differences.

-Background: Please explain briefly in one paragraph the impact of tumor and immune microenvironment in glioblastoma progression and treatment resistance.

Figure-1: please define the source of this figure. Please not that for including MRI images from patients you need to have an extra section for ethical approval. If this is the case, please include a separate section by adding the details. Usually in reviews we use illustrations, and this is what I would encourage the authors to include.

-Please check the full manuscript for typos and grammatical errors.

-It is highly recommended the authors to include an additional figure-illustrator showing the impact and interplay of immune and other cells within the TME of PBZ and the underlined mechanisms. Like a summary illustration of the cellular part.

-Figure 2: please remove the word “shows” from the figure legend. Again, here if you show images from patients, this need to be specified with an appropriate ethical approval section.

-Table 1: please remove abbreviations and include in the legend. Also, you should include an additional column with the highlighted references.

-An additional image showing the mechanisms relevant to the PBZ-mediated molecular and genomic/transcriptomics characteristics (as a summary illustration) is highly recommended.

- It would be better the authors to write a separate brief discussion session including the key concepts and be more critical summarizing the main findings and suggesting future research that will further improve current findings

Comments on the Quality of English Language

Minor edits are needed.

Author Response

REVIEWER 1

The manuscript entitled: “Current knowledge about peritumoral microenvironment in glioblastoma” by Trevisi and Mangiola, is a narrative review article that summarizes the most relevant findings about the biological characteristics of PBZ currently available under the medical perspective. This is an interesting study that fits within the scope of the journal and the special issue. I recommend this manuscript to be accepted for publication in the Cancers Journal, after some major revisions. Please find below the comments-suggestions and revisions that will help the authors improve the current version of this manuscript:

Major/minor comments:

Abstract:

-Results: since this is review article there is no need for this part in the abstract.

Authors: We amended accordingly.

-Lines 29-30: ” Understanding of PBZ could be relevant to develop more effective treatments to prevent GBM recurrence”: since you are talking about pre-malignant microenvironment it would be better to rephrase ”to prevent GBM development and recurrence”.

Authors: We amended accordingly.

- Background: It would be better to define the glioblastoma according to the new WHO2021 Classification at the beginning by briefly mentioning the different glioma grades in the first paragraph and explaining the GBM differences.

Authors: a detailed description of the new classification system has now been added to the very beginning of the new Introduction (previous Background) section.

- Background: Please explain briefly in one paragraph the impact of tumor and immune microenvironment in glioblastoma progression and treatment resistance.

Authors: a brief paragraph on the role of TME in GBM has been added in the Introduction.

- Figure-1: please define the source of this figure. Please not that for including MRI images from patients you need to have an extra section for ethical approval. If this is the case, please include a separate section by adding the details. Usually in reviews we use illustrations, and this is what I would encourage the authors to include.

Authors: We agree with the reviewer that it is unusual to use original neuroradiological imaging or surgical pictures in reviews. Nonetheless, we believe that showing our surgical planning and sampling better clarifies our definition of “PBZ”. Regarding Ethical Approval, both Fig.1 and Fig.2 were selected by patients who accepted to be enrolled in a prospective study on GBM authorized by Chieti‐Pescara Local Ethics Committee (E.C. number 08/21.05.2020). All patients gave written consent to the study and consented to the use of their radiological and biological data in anonymous form.

We included above details in the manuscript.

- Please check the full manuscript for typos and grammatical errors.

Authors: Thank you for the suggestion, we fully revised the manuscript to correct typos and grammatical errors.

-It is highly recommended the authors to include an additional figure-illustrator showing the impact and interplay of immune and other cells within the TME of PBZ and the underlined mechanisms. Like a summary illustration of the cellular part.

Authors: We added a new figure (Figure 3) as a summary illustration of the cellular part.

-Figure 2: please remove the word “shows” from the figure legend. Again, here if you show images from patients, this need to be specified with an appropriate ethical approval section.

Authors: There was an error in the order of text and figure, as the sentence was part of the manuscript and not of the figure legend. We corrected the wrong pagination.

-Table 1: please remove abbreviations and include in the legend. Also, you should include an additional column with the highlighted references.

Authors: We corrected the Table according to Reviewer’ suggestions.

-An additional image showing the mechanisms relevant to the PBZ-mediated molecular and genomic/transcriptomics characteristics (as a summary illustration) is highly recommended.

Authors: We added a new figure (Figure 3) as a summary illustration of the cellular part.

- It would be better the authors to write a separate brief discussion session including the key concepts and be more critical summarizing the main findings and suggesting future research that will further improve current findings

Authors: We included a discussion section, including our previous “ key concepts”,  and added a paragraph to summarize findings and to encourage future research focusing on PBZ molecular mechanisms, which, as shown in the manuscript, only partially overlap with those found in tumor core.

Reviewer 2 Report

Comments and Suggestions for Authors

This paper presents the most up-to-date understanding of the peritumoral area in glioblastoma. The text is clear, concise and intelligently summarizes several original research articles on the subject. The included summary table is also useful for readers.

Authors have experience on the topic and the authors have provided valuable insights into the tumor microenvironment of glioblastoma. However, a conclusion offering suggestions on how to utilize the data presented and initiate further research is needed. 

Author Response

REVIEWER 2

This paper presents the most up-to-date understanding of the peritumoral area in glioblastoma. The text is clear, concise and intelligently summarizes several original research articles on the subject. The included summary table is also useful for readers.

Authors have experience on the topic and the authors have provided valuable insights into the tumor microenvironment of glioblastoma. However, a conclusion offering suggestions on how to utilize the data presented and initiate further research is needed.

Authors: Thank you for appreciating our work. We included a paragraph to encourage future research focusing on PBZ molecular mechanisms, which, as shown in the manuscript, only partially overlap with those found in tumor core.

Round 2

Reviewer 1 Report

Comments and Suggestions for Authors

The manuscript "Current knowledge about peritumoral microenvironment in glioblastoma" is an interesting article and has been significantly improved after the revisions. There is one minor issue that needs to be solved before the acceptance for publication:

1) Please improve the quality of the images and especially the Figure 3 which is new figure.

2) Also the figure legend of Figure 3 is too long-please summarize this legend if possible.

Author Response

The manuscript "Current knowledge about peritumoral microenvironment in glioblastoma" is an interesting article and has been significantly improved after the revisions. There is one minor issue that needs to be solved before the acceptance for publication:

1) Please improve the quality of the images and especially the Figure 3 which is new figure.

R: We uploaded a new zip folder with high quality Figures and embedded such Figures in the manuscript

2) Also the figure legend of Figure 3 is too long-please summarize this legend if possible.

R: We shortened Figure 3 legend